# Medically Actionable Secondary Findings from Whole-Exome Sequencing (WES) Data in a Sample of 3972 Individuals

**DOI:** 10.3390/ijms26083509

**Published:** 2025-04-09

**Authors:** Mafalda Melo, Mariana Ribeiro, Paulo Filipe Silva, Susana Valente, Filipe Alves, Margarida Venâncio, Jorge Sequeiros, João Parente Freixo, Diana Antunes, Jorge Oliveira

**Affiliations:** 1Medical Genetics Unit, Hospital Dona Estefânia, Unidade Local de Saúde de Sao José, 1169-045 Lisbon, Portugaldiana.antunes@ulssjose.min-saude.pt (D.A.); 2Centre for Predictive and Preventive Genetics, Institute for Molecular and Cell Biology (CGPP-IBMC), 4200-135 Porto, Portugal; mpribeiro@i3s.up.pt (M.R.); paulo.silva@ibmc.up.pt (P.F.S.); svalente@i3s.up.pt (S.V.);; 3Instituto de Investigação e Inovação em Saúde (i3S), University of Porto, 4200-135 Porto, Portugal; 4ICBAS School of Medicine and Biomedical Sciences, University of Porto, 4050-313 Porto, Portugal; 5Unit for Multidisciplinary Research in Biomedicine (UMIB), ICBAS/ITR-Laboratory for Integrative and Translational Research in Population Health, University of Porto, 4050-313 Porto, Portugal; 6NOVA National School of Public Health (ENSP), NOVA University Lisbon, 1600-560 Lisbon, Portugal

**Keywords:** whole-exome sequencing, actionable secondary findings, genomic medicine

## Abstract

The application of whole-exome sequencing (WES) for diagnostic purposes has the potential to unravel secondary findings unrelated with the primary reason of testing. Some of those might be of high clinical utility and comprise disease-causing variants in genes, related to life-threatening and clinically actionable diseases. Clarifying the allelic frequencies of such variants in specific populations is a crucial step for the large-scale deployment of genomic medicine. We analysed medically relevant variants in the 81 genes from the American College of Medical Genetics and Genomics (ACMG) v3.2 list of actionable loci, using WES data from a diagnostic laboratory cohort of 3972 persons, tentatively resampled to represent the Portuguese population geographic distribution. We identified medically actionable variants in 6.2% of our cohort, distributed across several disease domains: cardiovascular disorders (3.0%), cancer predisposition (2.0%), miscellaneous disorders (1.1%), and metabolic disorders (0.1%). Additionally, we estimated a frequency of heterozygotes for recessive disease alleles of 11.1%. Overall, our results suggest that medically actionable findings can be identified in approximately 6.2% of persons from our population. This is the first study estimating medically actionable findings in Portugal. These results provide valuable insight for patients, healthcare providers, and policymakers involved in advancing genomic medicine at the national and international level.

## 1. Introduction

Whole-exome sequencing (WES) has become widely used in clinical practice to diagnose hereditary diseases. In addition to this primary purpose, WES holds a strong potential to uncover findings with significant health implications that may be unrelated to the original indication for testing. These findings can be unexpectedly discovered (incidental findings) or intentionally sought (secondary findings).

The American College of Medical Genetics and Genomics (ACMG) published guidelines for reporting secondary findings that could influence the medical management of both adults and minors undergoing genomic testing. These guidelines allow for opting out from receiving such findings. Over time, the ACMG has progressively expanded its list of genes recommended to identify secondary findings. In 2013, the list included 57 genes; these grew to 59 in 2017, 73 in 2021, 78 in 2022, and 81 genes by 2023. They were selected with specific criteria: association with highly penetrant diseases, long asymptomatic or prodromal stages, significant contribution to morbidity and/or mortality, availability of preventive measures or treatments, and potential for early detection to reduce long-term health risks effectively [1,2,3,4,5,6,7].

In contrast, the European Society of Human Genetics (ESHG) recommended a more cautious approach, aligned with the concept of opportunistic screening. The ESHG suggests further research and does not recommend a specific list of genes [8,9].

Since the introduction of these guidelines, various research groups have reported secondary findings identified in diverse populations. These efforts have contributed to understanding the clinical significance and implications of such findings in different genetic groups; they resorted to different sampling strategies [10,11,12,13,14,15,16,17,18,19,20,21,22,23,24,25,26,27,28,29,30,31,32,33,34,35,36,37,38,39].

A key aspect of identifying and classifying secondary findings relies on variant annotation and interpretation, which is significantly facilitated by curated databases such as ClinVar and HGMD (Human Gene Mutation Database). ClinVar is a publicly accessible, expert-curated database that aggregates information on the clinical significance of genetic variants, while HGMD compiles curated data on disease-causing variants reported in the literature. The integration of ClinVar and HGMD into variant-filtering pipelines enhances the reliability of pathogenicity classifications by incorporating collective knowledge from laboratories worldwide. Additionally, research efforts like ours contribute to enriching such databases by reporting novel or reclassified variants, ultimately improving the accuracy of variant interpretation for the broader scientific and clinical community.

In Portugal, however, the frequency of pathogenic (PAT) or likely pathogenic (L-PAT) variants in ACMG-actionable genes remained unknown, and no national policies for reporting secondary findings have been issued. Access to this information is crucial to support informed decision-making by patients, genetics laboratories, healthcare providers, and policymakers. Ultimately, it might support the development of health policies for genomic screening in each country.

In this study, we aimed at determining the frequency of PAT/L-PAT variants at loci listed as medically actionable by the ACMG (*n* = 81), and estimating the overall frequency of secondary findings in a diagnostic cohort at the Centre for Predictive and Preventive Genetics of the Institute for Molecular and Cellular Biology (CGPP-IBMC), which was resampled to be potentially representative of the Portuguese population.

## 2. Results

### 2.1. Dataset

A subset of 3972 persons was selected from a global dataset of 12,167 WES data, as described in the Material and Methods section, and data were anonymized.

### 2.2. Variant Filtering and Curation

In the selected cohort, there were 12,336 variants in the 81 ACMG medically actionable genes (Figure 1). To select variants of sufficient quality, we limited the analysis to those with a minimum read depth of 10× and genotype quality of 60. In general, each of the remaining 11,207 variants was then annotated with information available from (i) our internal laboratory database, (ii) ClinVar and HGMD, (iii) gnomAD, and (iv) molecular consequence predictors, by local bioinformatic analysis. After the filtering steps, 164 PAT/L-PAT variants automatically included or present in specific disease databases were included. Then, 5289 variants required further revision and manual curation using the ACMG/AMP guidelines and gene specific recommendations, when available; of these, an additional 95 variants met the inclusion criteria.

Thus, a total of 259 variants were classified as PAT/L-PAT, across 50 genes associated with autosomal dominant (*n* = 42), recessive (*n* = 7), and X-linked (*n* = 1) phenotypes (Appendix A).

### 2.3. Allele Counts

After the interpretation and classification of each of the 259 variants, we estimated the number of mutated alleles for each variant and corresponding gene. Table 1 provides a summary of the total number of variants per gene and their combined allele counts, distributed across the respective disease group.

As expected, mutated alleles were more frequent in genes related to recessive (*n* = 449), followed by dominant (*n* = 238), and then X-linked (*n* = 4) diseases.

### 2.4. Novel Variant Findings

Out of the 259 variants identified, we found 68 novel PAT/L-PAT variants (~26%) that have not been previously reported in the literature (as ascertained in the HGMD 2024.4 Pro and ClinVar databases and literature review). Detailed information on these novel variants is presented in Table 2. Their proportion varied across phenotype groups, with 30% in cancer-related genes, 38% in cardiovascular genes, and 6% in the miscellaneous group, while no novel findings were identified in the metabolic group.

### 2.5. Medically Actionable Findings and Carriers

The ACMG recommendations specify reporting only di-allelic (homozygous or compound heterozygous variants at the same locus) PAT/L-PAT variants in recessive disease genes. The frequency of potential medically actionable findings was estimated separately from carriers’ frequency, based on the zygosity of the variant, the number of mutated alleles, and the total number of alleles analysed.

#### 2.5.1. Frequency of Medically Actionable Findings

A total of 246 persons (6.19%) were estimated to have a potential medically actionable finding (Figure 2 and Appendix A) in the cohort, distributed across four disease groups: cardiovascular, cancer, metabolic, and miscellaneous disorders.

Actionable variants in genes for cardiovascular disease were the most frequent, present in 120 individuals, 48.78% of all persons with medically actionable findings and 3.02% of our cohort; these were subdivided into cardiomyopathies, CMPs (*n* = 61, 1.54%); dyslipidaemias (*n* = 23, 0.58%); rhythm disorders (*n* = 18, 0.45%); and hereditary connective tissue disorders, HCTDs (*n* = 18, 0.45%), as detailed in Appendix A. Among the CMPs, variants related to dilated CMP (DCM) were the most frequent (*n* = 30, 0.76%), found in *TTN*, *TNNT2*, *LMNA*, *FLNC*, *DES*, and *TNNC1*, followed by hypertrophic CMP (HCM) variants (*n* = 22, 0.55%), in *MYH7* and *MYBPC3*. Arrhythmogenic right ventricular CMP (ARVC) variants were the least common (*n* = 9, 0.23%), identified in *PKP2*, *DSP*, and *DSC2*. Among the dyslipidaemias, variants in *LDLR* (*n* = 19, 0.48%) and *APOB* (*n* = 4, 0.10%) were found, associated with familial hypercholesterolemia (FH). Arrhythmia predisposition variants were found in *KCNQ1* (*n* = 11, 0.28%), *KCNH2* (*n* = 2, 0.05%), and *SCN5A* (*n* = 5, 0.13%) genes. Finally, HCTD findings were reported in *FBN1* (*n* = 8, 0.20%), *MYH11* (*n* = 5, 0.13%), *TGFBR2* (*n* = 3, 0.08%), and *COL3A1* (*n* = 2, 0.05%).

Pathogenic variants in genes predisposing to hereditary cancer were present in 80 persons, or 32.52% of all with medically actionable findings, and 2.01% of the cohort (see Appendix A for details). The most frequent were associated with hereditary breast and ovarian cancer (HBOC) and Lynch syndrome/hereditary nonpolyposis colorectal cancer (HNPCC), in 25 (0.63%) and 24 (0.60%) persons, respectively. HBOC-related variants were present in *BRCA2* (*n* = 18, 0.45%), *PALB2* (*n* = 4, 0.10%), and *BRCA1* (*n* = 3, 0.08%), and HNPCC variants in *PMS2* (*n* = 11, 0.28%), *MSH6* (*n* = 9, 0.23%), and *MSH2* (*n* = 4, 0.10%). Less frequent cancer actionable variants were seen in *RET* (*n* = 6, 0.15%), *TSC2* (*n* = 6, 0.15%), *PTEN* (*n* = 4, 0.10%), *VHL* (*n* = 3, 0.08%), *BMPR1A* (*n* = 2, 0.05%), *MUTYH* (*n* = 2, 0.05%), RB1 (*n* = 2, 0.05%), *APC* (*n* = 1, 0.03%), *MEN1* (*n* = 1, 0.03%), *NF2* (*n* = 1, 0.03%), *SMAD4* (*n* = 1, 0.03%), *STK11* (*n* = 1, 0.03%), and *TP53* (*n* = 1, 0.03%) genes.

Concerning metabolic disorders, PAT/L-PAT variants were identified in four persons (three females and one male), in *GLA*, associated with Fabry disease, 1.63% of all with medically actionable findings and 0.10% of the cohort (refer to Appendix A).

In the group of miscellaneous disorders, PAT/L-PAT variants were identified in 42 persons, 17.07% of all with medically actionable findings and 1.06% of the cohort (outlined in Appendix A). Most represented variants for *TTR*-related hereditary amyloidosis (*n* = 17, 0.43%) and *RYR1*-related malignant hyperthermia (*n* = 13, 0.33%). Other, less frequent variants were related to *HNF1A*-maturity onset of diabetes of the young, MODY, (*n* = 7, 0.18%); *HFE*-related haemochromatosis (*n* = 2, 0.05%); *ACVRL1*-hereditary haemorrhagic telangiectasia (*n* = 2, 0.05%); and *ATP7B*–Wilson disease (*n* = 1, 0.03%).

#### 2.5.2. Carrier Frequencies

In addition to medically actionable alleles, we estimated that 439 persons (11.05%) in our cohort were carriers of PAT/L-PAT disease-causing alleles in recessive actionable disease genes (Figure 2 and Appendix A). These variants were most frequent in *HFE* (*n* = 207, 5.21%), *MUTYH* (*n* = 91, 2.29%) and *ATP7B* (*n* = 73, 1.84%), associated with hereditary haemochromatosis, *MUTYH*-associated polyposis (MAP), and Wilson disease. Less frequently, variants in heterozygosity were found in *RPE65* (*n* = 21, 0.53%), *GAA* (*n* = 20, 0.50%), *BTD* (*n* = 15, 0.38%), and *TRDN* (*n* = 12, 0.30%), associated with *RPE65*-related retinopathy, Pompe disease, biotinidase deficiency, and catecholaminergic polymorphic ventricular tachycardia. Further details are provided for each disorder group—cardiovascular, cancer, metabolic, and miscellaneous—in Appendix A, respectively.

## 3. Discussion

This research developed a strategy to obtain a potentially representative sample of the Portuguese population using WES-derived data from a diagnostic setting. The final purpose was to determine the frequency of PAT/L-PAT variants at ACMG’s medically actionable loci. Such data, however, were previous unavailable for our Portuguese population.

We could infer that 6.19% of persons had potentially actionable findings. This is in line with previous reports, ranging from 0.59% to 12.64% [10,11,12,13,14,15,16,17,18,19,20,21,22,23,24,25,26,27,28,29,30,31,32,33,34,35,36,37,38,39]. Several factors may explain this variation, including differences in populations, sample size, study design, the number of genes screened, sequencing technology, and variant filtering, interpretation, and classification, in addition to actionability criteria, and the increasing number of validated PAT/L-PAT variants in disease databases over time.

This study may also stand out from others for several reasons: (i) unlike most others, our cohort was resampled to be as representative as possible of the general population; (ii) the sample size was larger and more significant considering our population (of about 10.5 million people); (iii) the most recent ACMG list (v. 3.2, 2023) of 81 genes was analysed; and (iv) variant filtering and interpretation followed the most recent guidelines and gene specifications, and included a manual curation step.

In line with previous studies, PAT/L-PAT variants associated with “silent” but life-threatening diseases were identified within four groups: cardiovascular, cancer, miscellaneous, and metabolic disorders. Variants associated with cardiovascular diseases were the most frequent, and included cardiomyopathy, arrhythmias, dyslipidaemia, and connective tissue disorders.

CMP-related variant frequencies were aligned with similar studies or reported disease prevalence (DCM: 1:500; HCM: 1:500; ARVC: 1:1000 to 1:5000) and penetrance (DCM: 45%; HCM: 40%; ARVC: 30–75%) [40,41,42]. Likewise, primary arrhythmia variants, notably long QT syndrome, LQTS (0.45%), matched reported prevalence (1:2500) and penetrance (75%) [43]. Both CMPs and LQTS pose sudden death risks, warranting regular clinical surveillance, pharmacotherapy, and potentially implantable cardioverter defibrillators.

Concerning dyslipidaemia, the frequency of FH variants (0.58%) was in accordance with previously estimated disease frequency (1:200 to 1:250) in a meta-analysis [44]. FH leads to coronary heart disease events and death, if untreated. It has been estimated that only less than 10% of persons with FH are diagnosed, and even less receive treatment. Thus, early detection and possible treatment with lipid-lowering drugs are crucial.

On the other hand, HCTD-related findings, including Marfan syndrome (MS, 0.20%), familial thoracic aortic aneurism and dissection (FTAAD, 0.13%), Loeys–Dietz syndrome (LDS, 0.08%), and the vascular type of Ehlers–Danlos syndrome (vEDS, 0.05%) were more frequent than expected, according to disease frequency in the literature (1:3300 to 1:20,000 for MS; 1:5000 to 1:4,000,000 for FTAAD; unknown for LDS; and 1:50,000 for vEDS) [45,46,47,48]. Due to an increased risk of rupture of the aorta, early diagnosis is life-saving, in order to proceed with surveillance, therapy, and eventually prosthetic surgery.

The second most frequent group of findings was related with cancer, with HBOC and HNPCC being the most common. The frequency of reportable HBOC-related variants was estimated at 0.63%. This is 2–5-fold higher, when compared to the described frequency of *BRCA* pathogenic variants (1:400 to 1:800) [49]. However, this rate may be underestimated in our cohort, given the limitation to detect the NM_000059.4(*BRCA2*):c.156_157insAlu (a known founder variant in the Portuguese population) using WES data [50]. Mismatch-repair variants showed a frequency of 0.60%, similarly to the reported prevalence of HNPCC of 1:279 in the general population, and incomplete penetrance (>90% for colonic adenomas) [51]. All these findings support the relevance of an early diagnosis of hereditary cancer syndromes, since these persons may be offered clinical surveillance, following family history and knowing genetic status.

In the metabolic disorders group, we highlight the findings related to Fabry disease. Our screening showed a frequency of 0.10% for variants in *GLA*, corroborating recent data on the prevalence of Fabry disease (1:1250 to 1:5732) [52,53,54,55,56]. Specifically, NM_000169.3:c.337T>C, p.(Phe113Leu), a founder variant in the region of Guimarães, Portugal, and associated with the late-onset form, was found in 0.03% [57]. This higher prevalence of Fabry disease is related to knowledge of a broader phenotypic spectrum for *GLA*, encompasses the classical and a late-onset form. Persons with this late-onset phenotype may have significant diagnostic delay [58]. Genetic testing may allow for earlier diagnosis, critical for the efficiency of enzyme replacement or chaperone therapy, recommended to be initiated as early as possible.

*TTR*-related hereditary amyloidosis has a significant cluster in northern Portugal, the largest worldwide [59,60]. In our data, the frequency of *TTR* pathogenic variants was 0.43%. The p.Val50Met variant (NM_000371.4:c.148G>A) showed a frequency of 0.28%, comparable to that reported in northern Portugal (1:538) [61]. This variant is predominantly associated with *TTR*-related amyloid neuropathy [55]. Surprisingly, the p.Val142Ile variant (NM_000371.4:c.424G>A), known to be associated with the *TTR*-related cardiac amyloidosis, particularly prevalent in persons of African ancestry, was also present in 0.16% [62]. The higher prevalence of *TTR* variants identified supports the greater awareness for *TTR* amyloidosis in Portugal, allowing for early diagnosis, follow-up, and treatment, including liver transplant, pharmacotherapy, and an implantable cardioverter-defibrillator when indicated.

We also identified variants in the *RYR1* gene with an overall frequency of 0.33%. The exact worldwide prevalence of malignant hyperthermia susceptibility (MHS) has been difficult to clarify given disparities in clinical diagnosis methods and criteria. Using population genomics data, the estimated prevalence of an MHS-related pathogenic variant was 1:1450 to 1:1556 [63,64]. Here, we report a frequency 10 times higher, after having restricted our investigation to variants functionally validated and curated by the European Malignant Hyperthermia Group. The assessment of MHS risk is relevant so that effective measures can be put in place, such as the avoidance of certain anaesthetics upon surgery.

In addition, we identified heterozygous variants in genes associated with autosomal recessive diseases. In total, 11.05% of persons were carriers of any high-risk ACMG actionable genes linked to recessive diseases, including *HFE*, *MUTYH*, *ATP7B*, *RPE65*, *GAA*, *BTD*, and *TRDN*. In our cohort, the frequency of heterozygotes at *HFE* (5.21%), *MUTYH* (2.29%), *ATP7B* (1.83%), *GAA* (0.50%), and *BTD* (0.38%) was comparable to previous estimates in the general population (1:10 for *HFE*, 1:50 for *MUTYH*, 1:50 to 1:90 for *ATP7B*, 1:70 for *GAA*, and 1:120 for *BTD*) [51,65,66,67,68,69,70,71]. For comparison, the American College of Obstetricians and Gynecologists endorsed carrier screening for cystic fibrosis and spinal muscular atrophy, as well as other diseases with a carrier frequency ≥1/100 [72]. Our data may provide support as to which genes should be selected for preconception carrier screening based on population carrier frequency, to address together with considerations on penetrance, severity, and predictable genotype–phenotype correlation. Our work has some limitations, most of which are transversal to similar published studies:Lab cohort bias. Our sample was derived from cases ascertained for the genetic diagnosis of various Mendelian disorders; therefore, a few persons in the cohort may already be affected by a disease attributable to one of the genes in the ACMG list. Despite this, when excluding L-PAT/PAT variants listed as primary diagnosis in the genetic test reports of these patients, the overall frequency did not differ significantly (only 0.6%).Gene list. We limited our analysis to the current set of the ACMG genes. We did not consider other clinically relevant genes, such as those curated by the ClinGen Actionability Working Group, for instance. The inclusion of additional conditions, some of specific impact in the Portuguese population, should be considered in future studies, which might increase the overall frequency of actionable findings.Study design. In order to minimise the impact of data used, our project protocol prevented us from including individual-level information regarding ethnic background, age, gender, reason for referral for WES, or phenotype. Additionally, the genotypes obtained were related to the whole cohort, not the patient. Consequently, we were not able to estimate compound heterozygosity or the number of findings per individual.Technical limitations. The methodologies used may have led to missed variants due to (i) the intrinsic WES limitation to detect deep intronic, triplet repeat expansion, and structural variants; (ii) the use of different capture kits along time within this cohort; (iii) incomplete coverage in some regions; (iv) not considering structural variants, including copy number variants (CNVs); and (v) global minor allele frequency (MAF) cut-off.Pseudogenes or highly homologous genomic regions. Several ACMG genes, such as BMPR1A, BRCA1, CALM1, FLNC, PKP2, PMS2, PTEN, and TTN, are associated with pseudogenes or highly homologous genomic regions, which can compromise the accurate detection of variants. The presence of pseudogenes may lead to false negatives due to challenges in aligning reads from functional genes and pseudogenes, potentially missing disease-causing variants. This aspect should be considered when interpreting our findings.Potential for false-positive interpretation of variants. Variants accurately classified as PAT/L-PAT, based on available evidence, may not, in fact, be disease-causing, due to incomplete penetrance or variable expressivity. This is exacerbated when genetic testing is performed in the context of population screening.Actionability. The term “actionable” is highly subjective and its application may fluctuate. The ClinGen Actionability Working Group is addressing this issue by curating the actionability of several gene-disease groups, including those listed by the ACMG. We took this into consideration; however, some gene-disease groups are not yet curated, and others are classified as actionable depending on individual-level information, such as age and sex, which were not considered due to our study protocol.

Future studies, including more persons, are needed to determine more accurately the frequency of ultra-rare disease-causing alleles in our study population. Our methodology could also be used to evaluate the allelic frequency for non-ACMG actionable loci. This could include other diseases known to have impact in Portugal, such as haemoglobinopathies, congenital adrenal hyperplasia, and cystic fibrosis; some of those genes may be of interest for the design of preconceptional screening programmes. Optimising guidelines for variant interpretation and reporting criteria in asymptomatic persons is also suggested, as well as research on long-term phenotypic effects of presumed PAT/L-PAT variants in the general population. Finally, it is recommended to perform responsible research on genomic screening, before considering its eventual implementation within healthcare services.

## 4. Materials and Methods

### 4.1. Study Design and Dataset of Exomes

This is a cross-sectional observational study with a quantitative approach. We analysed the WES data obtained from patients tested at CGPP-IBMC (12,167 samples, retrieved on 19 May 2023), in an anonymized and aggregated form. All patients had been clinically diagnosed or had a diagnostic suspicion of a disease for which the molecular test requested by their physician was either a virtual gene-panel based on WES or a complete WES analysis.

The WES raw data analysed in this study were generated through DNA fragmentation and capture using different kits, and subsequently massive parallel sequencing with Illumina’s short-read technology, specifically employing the HiSeq 2000, 2500, 4000 or the NovaSeq 6000 platforms (Illumina Inc, San Diego, CA, USA) over the years. Due to the evolving nature of sequencing technologies, the WES data were generated using different capture kits. Initially, sequencing was performed using the SureSelect Human All Exon V6 kit (Agilent Technologies, Santa Clara, CA, USA), later the Twist Human Core kit (Twist Biosciences, San Francisco, CA, USA) was used, and now the Twist Human Comprehensive (Twist Human Core Exome + Human RefSeq Panel) kit is being applied. Using an in-house validated and accredited bioinformatic diagnostics pipeline, reads were aligned with the human genome reference sequence GRCh37 using the Burrows-Wheeler Aligner (BWA v0.7.15-r1140), followed by variant calling with the Broad Institute’s Genome Analysis Toolkit (GATK v3.7-0-gcfedb67) [73] and variant annotation using VarSeq software (v2.6.2, Golden Helix Inc, Bozeman, MT, USA) and in-house scripts.

### 4.2. Resampling from CGPP-IBMC Clinical Database

The methodology used to generate a sample approaching representativeness of the Portuguese population is detailed elsewhere [74]. Briefly, to remove possible population biases, the CGPP-IBMC cohort was resampled to (i) include only one sample per family; (ii) include one partner per consanguineous couple; and (iii) remove pre-natal or foetal samples. The final subset used was corrected for the geographic distribution of cases and population size, by using data from Census 2021 [75]; also, distinct datasets were made according to the known distribution of the population, by municipalities both in mainland Portugal, and Madeira and the Azores archipelagos. For the purposes of this study, a unique VCF file containing all variants from the 3972 samples was generated and used (process illustrated in Appendix A), representing the entire population of Portugal.

### 4.3. Selection of Genes for Which Reporting of Secondary Findings Is Recommended

The list of genes in the present analysis covered the 81 genes that are part of the revised guidelines on secondary findings from the ACMG, version 3.2 [7]. This list encompasses genes related to cancer (*n* = 28), cardiovascular (*n* = 40), inborn errors of metabolism (*n* = 4), and miscellaneous (*n* = 9) phenotypes (Appendix A).

### 4.4. Data Processing

All variants from the 3972 single-sample VCFs that did not pass the bioinformatics quality filters—specifically, those with read depth (RD) <10× and genotype quality (GQ) <60—were excluded (see Appendix A). The filtered, individual VCF files were then aggregated into a single multi-sample VCF file, as described in Appendix A. A BED file, containing genomic coordinates corresponding to all 81 ACMG genes (exonic coding regions plus 20 bp intronic flanks), was intersected with the full WES multi-sample VCF to select variants only in those genes (Appendix A). This multi-sample VCF was then further processed by using bcftools to calculate aggregate numbers such as allele count, allele number, allele frequency, and the number of homozygotes, heterozygotes, and hemizygotes, and to remove individual genotype information to ensure the irreversible anonymization of patients’ data (Appendix A). This aggregation and anonymization was performed for (i) all samples and (ii) resampled data.

### 4.5. Variant Annotation

Variants were systematically annotated with information available from (i) our in-house database, (ii) disease databases, namely, ClinVar (retrieved 5 December 2024) and HGMD (version Pro 2024.4, Qiagen Gmbh, Hilden, Germany), (iii) the population database gnomAD (versions 2.1.1, 3.1.2, and 4.1), and (iv) molecular consequence predictors.

### 4.6. Variant Filtering

The overall filtering and manual curation workflow are explained in Figure 1 and Appendix A. First, as mentioned in Section 4.4, variants were filtered based on confidence metrics including genotype quality (GQ, ≥60) and read depth (RD, ≥10×). Second, variants were prioritised according to prior classification in our in-house laboratory database, being automatically included if classified as PAT/L-PAT. Third, variants were filtered based on previous ClinVar classification, being included if classified as PAT/L-PAT 2 stars and excluded if BEN/L-BEN 2 stars. Fourth, variants were filtered based on their MAF, with the exclusion of variants with MAF ≥ 0.5% in gnomAD, except for the *HFE* variant NM_000410.4:c.845G>A (p.Cys282Tyr). Finally, variants were filtered according to their variant effect prediction (VEP), which is based on Sequence Ontology terms; variants with low predicted impact were excluded.

### 4.7. Manual Variant Curation, Classification, and Actionability

We further curated manually the remaining variants, following the guidelines for the interpretation and classification of sequence variants from the ACMG/AMP [76]. Additionally, specifications for variant classification of the Association for Clinical Genomic Science (ACGS), Clinical Genome Resource (ClinGen), Cancer Variant Interpretation Group UK (CanVIG-UK), and the European Molecular Genetics Quality Network (EMQN) were considered, when applicable (Appendix A) [77,78,79,80]. All variants classified as PAT/L-PAT, referring to the potential of the variant to cause a relevant phenotype according to the ACMG v3.2 recommendations, were considered to be medically actionable.

### 4.8. Calculation of Frequency of Actionable Findings

The overall frequency of actionable findings was then calculated considering the total size of the WES cohort and the number of genes.

## 5. Conclusions

To the best of our knowledge, this is the first evaluation of medically actionable findings in the Portuguese population, making this study a pioneering effort. We demonstrate that this population, as others, is expected to harbour medically actionable variants that can be identified through WES in routine diagnosis; however, while our study provides important insight, additional evidence is needed to take its findings to the population level.

This foundational research establishes a basis for future investigations into genomic screening and highlights the potential for data from WES-based genetic testing to significantly impact public health.

## Figures and Tables

**Figure 1 ijms-26-03509-f001:**
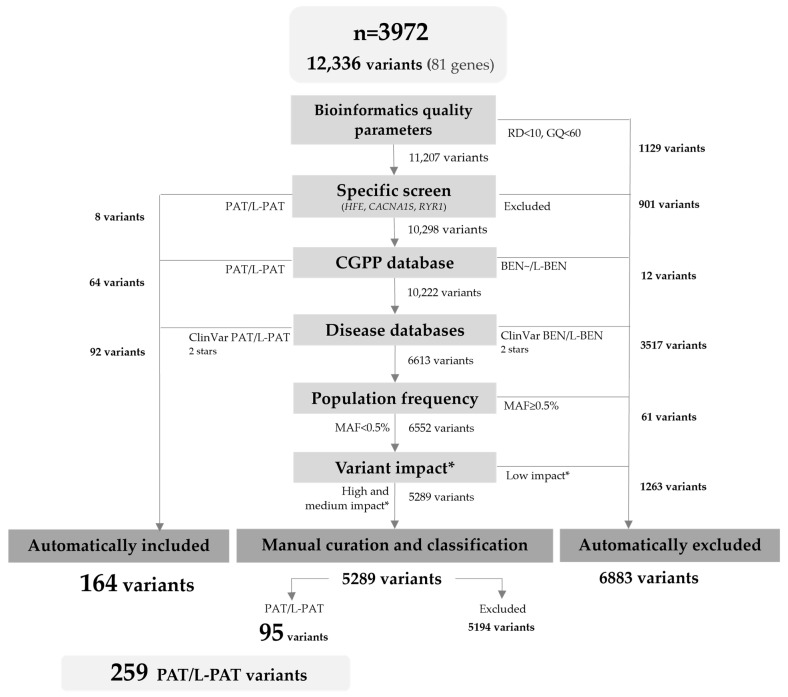
Flowchart of the variant filtering and curation process. RD: read depth; GQ: genotype quality; MAF: minor allele frequency of global population; PAT: pathogenic; L-PAT: likely pathogenic; BEN: benign; L-BEN: likely benign. * Variant impact according to Variant Effect Predictor (VEP) based on Sequence Ontology terms.

**Figure 2 ijms-26-03509-f002:**
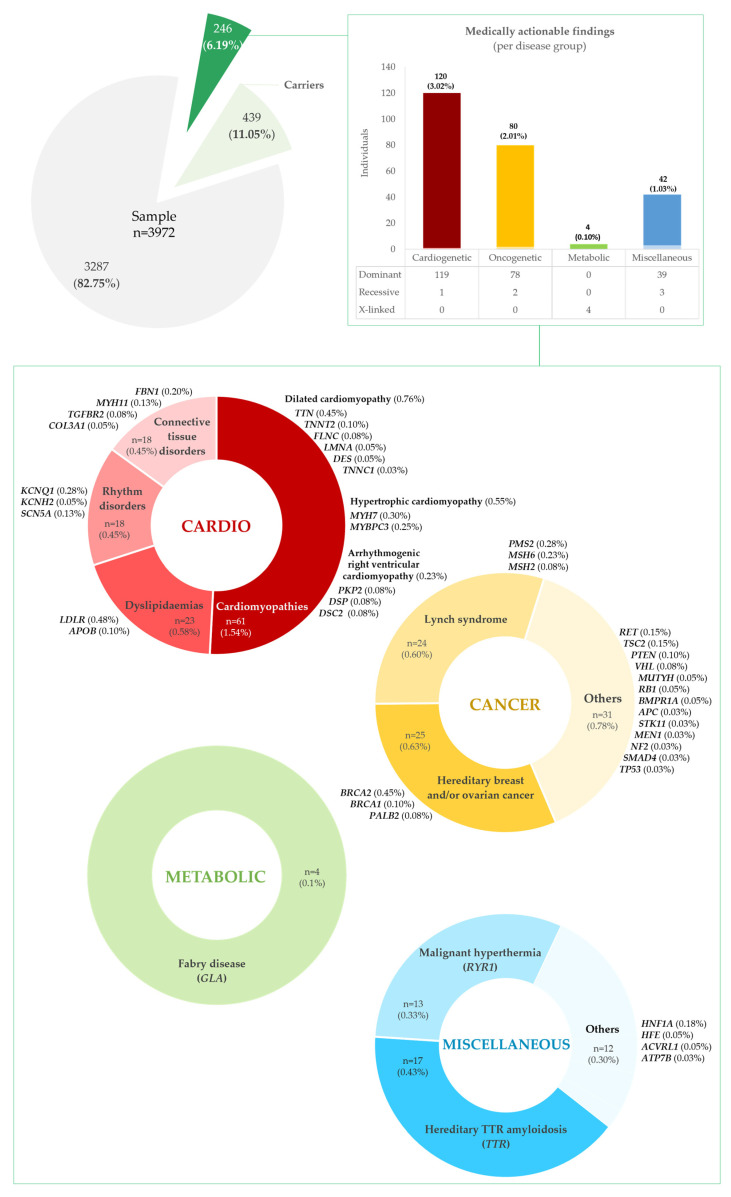
Total medically actionable findings and non-medically actionable findings (carrier status) present in our cohort of 3972 persons, presumed to be representative of the Portuguese population.

**Table 1 ijms-26-03509-t001:** Summary of genes by phenotype group with the number of pathogenic and likely pathogenic variants identified in our cohort according to ACMG/AMP criteria.

Disease	Gene	Disease Inheritance	No. of Variants Per Gene	Allele Count			Allelic Frequency (%)
				Total	Hom.	Het.	
Cancer phenotype group							
Familial adenomatous polyposis	*APC*	AD	1	1	0	1	0.013
Familial medullary thyroid cancer	*RET*	AD	3	6	0	6	0.076
Hereditary breast and/or ovarian cancer	*BRCA1*	AD	3	3	0	3	0.038
	*BRCA2*	AD	16	18	0	18	0.227
	*PALB2*	AD	2	4	0	4	0.050
Hereditary paraganglioma-pheochromocytoma syndrome	*SDHD*	AD	0	0	0	0	0.000
	*SDHAF2*	AD	0	0	0	0	0.000
	*SDHC*	AD	0	0	0	0	0.000
	*SDHB*	AD	0	0	0	0	0.000
	*MAX*	AD	0	0	0	0	0.000
	*TMEM127*	AD	0	0	0	0	0.000
Juvenile polyposis syndrome	*BMPR1A*	AD	2	2	0	2	0.025
Juvenile polyposis syndrome/hereditary haemorrhagic telangiectasia syndrome	*SMAD4*	AD	1	1	0	1	0.013
Li–Fraumeni syndrome	*TP53*	AD	1	1	0	1	0.013
Lynch syndrome/hereditarynonpolyposis colorectal cancer	*MLH1*	AD	0	0	0	0	0.000
	*MSH2*	AD	3	4	0	4	0.050
	*MSH6*	AD	8	9	0	9	0.113
	*PMS2*	AD	6	11	0	11	0.138
Multiple endocrine neoplasia type 1	*MEN1*	AD	1	1	0	1	0.013
MUTYH-associated polyposis	*MUTYH*	AR	17	95	4	91	1.196
Neurofibromatosis type 2	*NF2*	AD	1	1	0	1	0.013
Peutz–Jeghers syndrome	*STK11*	AD	1	1	0	1	0.013
PTEN hamartoma tumour syndrome	*PTEN*	AD	4	4	0	4	0.050
Retinoblastoma	*RB1*	AD	2	2	0	2	0.025
Tuberous sclerosis complex	*TSC1*	AD	0	0	0	0	0.000
	*TSC2*	AD	6	6	0	6	0.076
von Hippel–Lindau syndrome	*VHL*	AD	1	3	0	3	0.038
WT1-related Wilms tumour	*WT1*	AD	0	0	0	0	0.000
Cardiovascular phenotype group							
Aortopathies	*FBN1*	AD	8	8	0	8	0.101
	*TGFBR1*	AD	0	0	0	0	0.000
	*TGFBR2*	AD	2	3	0	3	0.038
	*SMAD3*	AD	0	0	0	0	0.000
	*ACTA2*	AD	0	0	0	0	0.000
	*MYH11*	AD	3	5	0	5	0.063
Arrhythmogenic right ventricular cardiomyopathy (a subcategory of arrhythmogenic cardiomyopathy)	*PKP2*	AD	3	3	0	3	0.038
	*DSP*	AD	3	3	0	3	0.038
	*DSC2*	AD	3	3	0	3	0.038
	*TMEM43*	AD	0	0	0	0	0.000
	*DSG2*	AD	0	0	0	0	0.000
Catecholaminergic polymorphic ventricular tachycardia	*RYR2*	AD	0	0	0	0	0.000
	*CASQ2*	AR	0	0	0	0	0.000
	*TRDN*	AR	6	12	0	12	0.151
Dilated cardiomyopathy	*TNNT2*	AD	3	4	0	4	0.050
	*LMNA*	AD	2	2	0	2	0.025
	*FLNC*	AD	3	3	0	3	0.038
	*TTN*	AD	18	18	0	18	0.227
	*BAG3*	AD	0	0	0	0	0.000
	*DES*	AD	2	2	0	2	0.025
	*RBM20*	AD	0	0	0	0	0.000
	*TNNC1*	AD	1	1	0	1	0.013
Ehlers–Danlos syndrome. vascular type	*COL3A1*	AD	2	2	0	2	0.025
Familial hypercholesterolemia	*LDLR*	AD	15	19	0	19	0.239
	*APOB*	AD	4	4	0	4	0.050
	*PCSK9*	AD	0	0	0	0	0.000
Hypertrophic cardiomyopathy	*MYH7*	AD	8	12	0	12	0.151
	*MYBPC3*	AD	10	10	0	10	0.126
	*TNNI3*	AD	0	0	0	0	0.000
	*TPM1*	AD	0	0	0	0	0.000
	*MYL3*	AD	0	0	0	0	0.000
	*ACTC1*	AD	0	0	0	0	0.000
	*PRKAG2*	AD	0	0	0	0	0.000
	*MYL2*	AD	0	0	0	0	0.000
Cardiovascular phenotype group							
Long QT syndrome types 1 and 2	*KCNQ1*	AD	6	12	2	10	0.151
	*KCNH2*	AD	2	2	0	2	0.025
Long QT syndrome 3. Brugada syndrome	*SCN5A*	AD	4	5	0	5	0.063
Long QT syndrome types 14–16	*CALM1*	AD	0	0	0	0	0.000
	*CALM2*	AD	0	0	0	0	0.000
	*CALM3*	AD	0	0	0	0	0.000
Inborn errors of metabolism phenotype group							
Biotinidase deficiency	BTD	AR	10	15	0	15	0.189
Fabry disease	GLA	XL	3	4	1	3	0.063
Pompe disease	GAA	AR	9	20	0	20	0.252
Ornithine transcarbamylase deficiency	OTC	XL	0	0	0	0	0.000
Miscellaneous phenotype group							
Hereditary haemochromatosis	*HFE*	AR	1	211	4	207	2.656
Hereditary haemorrhagic telangiectasia	*ACVRL1*	AD	2	2	0	2	0.025
	*ENG*	AD	0	0	0	0	0.000
Malignant hyperthermia	*RYR1*	AD	7	13	0	13	0.164
	*CACNA1S*	AD	0	0	0	0	0.000
Maturity-onset of diabetes of the young	*HNF1A*	AD	3	7	0	7	0.088
RPE65-related retinopathy	*RPE65*	AR	11	21	0	21	0.264
Wilson disease	*ATP7B*	AR	24	75	2	73	0.944
Hereditary TTR-related amyloidosis	*TTR*	AD	2	17	0	17	0.214

Note: Hom.—Homozygous; Het.—Heterozygous. AD: Autosomal dominant; AR: autosomal recessive; XL: X-linked.

**Table 2 ijms-26-03509-t002:** Novel pathogenic and likely pathogenic variants identified in this study that have not been previously reported in the literature.

Gene	cDNA (HGVS)	Predicted Splicing Impact	Protein Change (HGVS)	Freq. (gnomAD4.1) (%)	ClinVar ID (2 January 2025)
*BMPR1A*	NM_004329.3:c.231-2A>T	Y	-	-	2866138
	NM_004329.3:c.231-1G>T	Y	-	-	567998
*BRCA1*	NM_007294.4:c.109A>G	N	NP_009225.1:p.Thr37Ala	-	868146
*BRCA2*	NM_000059.4:c.2974A>T	N	NP_000050.3:p.Lys992*	-	-
	NM_000059.4:c.4933A>T	N	NP_000050.3:p.Lys1645*	-	51744
	NM_000059.4:c.7258delG	N	NP_000050.3:p.Glu2420Asnfs*47	-	-
*MEN1*	NM_130799.3:c.467G>C	N	NP_570711.2:p.Gly156Ala	-	-
*MSH2*	NM_000251.3:c.2084T>G	N	NP_000242.1:p.Val695Gly	-	-
*MSH6*	NM_000179.3:c.195_199delACCGC	N	NP_000170.1:p.Pro66Glnfs*22	0.0001	-
	NM_000179.3:c.198_199insTT	N	NP_000170.1:p.Pro67Phefs*15	-	-
	NM_000179.3:c.841G>T	N	NP_000170.1:p.Gly281*	-	2673649
	NM_000179.3:c.2437A>T	N	NP_000170.1:p.Lys813*	0.0001	1791241
	NM_000179.3:c.3682_3698del	N	NP_000170.1:p.Ala1228Argfs*4	-	-
*MUTYH*	NM_001128425.2:c.788+2_788+4delTAG	Y	-	-	-
	NM_001128425.2:c.785_786insG	N	NP_001121897.1:p.Trp263Leufs*66	-	-
	NM_001128425.2:c.781delC	N	NP_001121897.1:p.Gln261Serfs*5	-	-
*PTEN*	NM_000314.8:c.802-1_805delGGACA	N	NP_000305.3:p.?	-	-
	NM_000314.8:c.804_805insTTTTT	N	NP_000305.3:p.Lys269Phefs*9	-	-
*RB1*	NM_000321.3:c.1422-2A>T	Y	-	0.0004	-
*SMAD4*	NM_005359.6:c.904+1_904+2ins(45)	Y	-	0.0053	-
*TSC2*	NM_000548.5:c.264_265delGT	N	NP_000539.2:p.Leu89Alafs*36	0.0160	45485999
	NM_000548.5:c.340G>T	N	NP_000539.2:p.Glu114*	-	65033
	NM_000548.5:c.775-2A>C	Y	-	-	-
	NM_000548.5:c.2340_2341ins(37)	N	NP_000539.2:p.Asp781Phefs*12	-	-
*APOB*	NM_000384.3:c.9743_9744insG	N	NP_000375.3:p.Ile3248Metfs*12	-	-
	NM_000384.3:c.9735delC	N	NP_000375.3:p.Gln3247Lysfs*19	-	-
	NM_000384.3:c.2297_2298delAA	N	NP_000375.3:p.Lys766Ilefs*25	-	1553385715
*COL3A1*	NM_000090.4:c.1429G>A	N	NP_000081.2:p.Gly477Arg	-	-
	NM_000090.4:c.2229+1G>A	Y	-	-	640856
*DES*	NM_001927.4:c.75_76insAG	N	NP_001918.3:p.Leu26Serfs*6	-	-
*DSC2*	NM_024422.6:c.1044_1047dupAAAT	N	NP_077740.1:p.Asp350Lysfs*2	-	-
	NM_024422.6:c.631-1G>A	Y	-	0.0001	2775190
*DSP*	NM_004415.4:c.107delG	N	NP_004406.2:p.Gly36Alafs*12	-	-
	NM_004415.4:c.1258G>T	N	NP_004406.2:p.Glu420*	-	-
	NM_004415.4:c.2572delG	N	NP_004406.2:p.Glu858Lysfs*6	-	-
*FBN1*	NM_000138.5:c.4282C>T	N	NP_000129.3:p.Arg1428Cys	0.0007	-
	NM_000138.5:c.4015_4016insTG	N	NP_000129.3:p.Cys1339Leufs*75	-	-
*FLNC*	NM_001458.5:c.502delT	N	NP_001449.3:p.Trp168Glyfs*84	-	-
	NM_001458.5:c.2550+1G>A	Y	-	-	-
*KCNH2*	NM_000238.4:c.1621C>T	N	NP_000229.1:p.Arg541Cys	0.0004	937094
*LDLR*	NM_000527.5:c.1315A>T	N	NP_000518.1:p.Asn439Tyr	-	375813
*MYBPC3*	NM_000256.3:c.2995-2A>G	Y	-	0.0001	-
*MYH7*	NM_000257.4:c.1756G>A	N	NP_000248.2:p.Val586Met	0.0004	1172186
*PKP2*	NM_004572.4:c.1489C>T	N	NP_004563.2:p.Arg497*	0.0003	78974
	NM_004572.4:c.328delA	N	NP_004563.2:p.Met110Cysfs*2	-	-
*SCN5A*	NM_198056.3:c.5306C>T	N	NP_932173.1:p.Ala1769Val	0.0001	-
*TGFBR2*	NM_003242.6:c.760C>T	N	NP_001020018.1:p.Arg279Cys	0.0001	213942
*TNNT2*	NM_001276345.2:c.87_88delGG	N	NP_001263274.1:p.Asp30Argfs*13	-	-
	NM_001276345.2:c.80G>A	N	NP_001263274.1:p.Trp27*	-	-
*TRDN*	NM_006073.4:c.1831+1G>A	Y	-	-	-
	NM_006073.4:c.1155delA	N	NP_006064.2:p.Lys385Asnfs*5	0.0013	-
	NM_001256021.2:c.601_610delCTGGCGAAAG	N	NP_001242950.1:p.Leu201Asnfs*19	0.0029	-
	NM_001256021.2:c.439_440delAA	N	NP_001242950.1:p.Lys147Aspfs*2	0.0001	2114339116
*TTN*	NM_001267550.2:c.107409_107410insCC	N	NP_001254479.2:p.Leu35804Profs*2	-	-
	NM_001267550.2:c.97573_97574insTC	N	NP_001254479.2:p.Asp32525Valfs*8	-	-
	NM_001267550.2:c.95576_95577delAA	N	NP_001254479.2:p.Lys31859Argfs*6	-	-
	NM_001267550.2:c.93623_93626dupAGCC	N	NP_001254479.2:p.Gln31210Alafs*8	-	-
	NM_001267550.2:c.84525G>A	N	NP_001254479.2:p.Trp28175*	-	-
	NM_001267550.2:c.79811dupT	N	NP_001254479.2:p.Arg26605Lysfs*19	-	-
	NM_001267550.2:c.70971_70972insT	N	NP_001254479.2:p.Leu23658Serfs*18	-	-
	NM_001267550.2:c.64266delA	N	NP_001254479.2:p.Asp21423Ilefs*2	0.0001	-
	NM_001267550.2:c.58709C>G	N	NP_001254479.2:p.Ser19570*	-	-
	NM_001267550.2:c.52975_52976delCA	N	NP_001254479.2:p.Gln17659Thrfs*6	0.0001	-
	NM_001267550.2:c.41845dupA	N	NP_001254479.2:p.Ile13949Asnfs*2	-	-
	NM_001267550.2:c.13184delT	N	NP_001254479.2:p.Leu4395Argfs*25	-	-
*ACVRL1*	NM_000020.3:c.830C>T	N	NP_000011.2:p.Thr277Met	0.0004	2731545
*ATP7B*	NM_000053.4:c.3959G>C	N	NP_000044.2:p.Arg1320Thr	0.0010	1479012
*RPE65*	NM_000329.3:c.1544G>A	N	NP_000320.1:p.Arg515Gln	0.0015	1052287

Footnote: Freq.—Frequency; Y—Yes; N—No. (end of table).

## Data Availability

The NGS datasets used for this article resulted from a clinical diagnostic setting and are not publicly available. The main data needed to assess the conclusions in the paper are presented in the main text or in the Appendix A.

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
