# Peer review of "Medically Actionable Secondary Findings from Whole-Exome Sequencing (WES) Data in a Sample of 3972 Individuals"

_ijms, 2025, doi:10.3390/ijms26083509_

Round 1
Reviewer 1 Report
Comments and Suggestions for Authors
Summary
Authors provided a comprehensive evaluation of the secondary findings output of the WES analysis.
Introduction
Very concise and interesting. Nevertheless, I suggest introducing the databases such as ClinVar that are pivotal for assigning the pathogenic or likely pathogenic description of the secondary findings individuated. Additionally, I recommend emphasizing the modern tools used for vcf filtration and variant prioritization. This process, in fact, is crucial for determining the selected variants (main and secondary findings).
Materials and methods
Subsection 4.1.
Could You provide additional informations regarding the of the suspected diseases for which WES has been proposed?
Reviewer 2 Report
Comments and Suggestions for Authors
see attached.
